# Exploring the mechanism of hyperpermeability following glycocalyx degradation: Beyond the glycocalyx as a structural barrier

Kyoko Abe[1], Junichi Tanaka[2], Kenji Mishima[2], Takehiko Iijima[1]*

1 Division of Anesthesiology, Department of Perioperative Medicine, School of Dentistry, Showa University, Ota City, Tokyo, Japan, 2 Division of Pathology, Department of Oral Diagnostic Sciences, School of Dentistry, Showa University, Shinagawa, Tokyo, Japan

* iijima@dent.showa-u.ac.jp

**Data Availability Statement:** All relevant data are within the paper and its Supporting Information files.

## Abstract

Pathological hyperpermeability is a morbidity involved in various systemic diseases, including sepsis. The endothelial glycocalyx layer (GCX) plays a key role in controlling vascular permeability and could be a useful therapeutic target. The purpose of the present study was to analyze the functional role of the GCX in vascular permeability and to elucidate its role in pathological conditions. First, male C57BL/6J wild-type mice were used as *in vivo* models to study the effects of sepsis and the pharmacological digestion of glycosaminoglycans (GAGs) on the GCX. Vascular permeability was evaluated using fluorescein isothiocyanate (FITC)-labeled dextran. Second, the changes in gene expression in vascular endothelial cells after GAGs digestion were compared between a control and a septic model using RNA sequencing. In the *in vivo* study, the glycocalyx was depleted in both the septic model and the group with pharmacological GAGs digestion. FITC-labeled dextran had leaked into the interstitium in the septic group, but not in the other groups. In the *in vitro* study, histamine decreased the transendothelial electrical resistance (TEER), indicating an increase in permeability. GAGs digestion alone did not change the TEER, and the effect of histamine on the TEER was not enhanced by GAGs digestion. The gene expression profiles after GAGs digestion differed from the control condition, indicating the initiation of signal transduction. In conclusion, we demonstrated that the structural barrier of the GCX does not solely determine the fluid permeability of the endothelial layer, since enzymatic depletion of the GCX did not increase the permeability. The gene expression findings suggest that the digestion of GAGs alone did not induce hyperpermeability either *in vitro* or *in vivo*, although sepsis did induce hyperpermeability. While GAGs degradation by itself does not appear to induce hyperpermeability, it may play an important role in initiating signal transductions.

**Funding:** We declare that our study was supported by Grants-in-Aid for Scientific Research [Grant 392 16K11762] from the Ministry of Education, Culture, Sports, Science and Technology of Japan.

**Competing interests:** The authors have declared that no competing interests exist.

## Introduction

Fluid management difficulties can be attributed to an intractable dysregulation of vascular permeability. The pathological mechanism responsible for permeability is being explored but is far from being completely understood. The vascular lumen is lined by the endothelial surface layer (ESL), which insulates the intraluminal fluid from the interstitial space and keeps the corpuscular blood elements and macromolecules at a physical distance from the apical surface of the endothelial cells [1–3]. The disruption of the ESL loosens this tight seal [4] and may lead to the initial step in the pathological entity of sepsis. This morphological change in the ESL can explain the hyperpermeability that occurs during sepsis [5–7]. Thus, the biological mechanism responsible for the induction of hyperpermeability should be further explored. The ESL consists of glycosaminoglycans (GAGs) and proteoglycans (PGs), together known as the glycocalyx, attached to albumin [8–10]. The glycocalyx covers the endothelial surface, upon which various receptors are formed; these receptors transmit signals into the endothelial cells [11]. Covering these receptors inhibits receptor activation and signal transduction, contributing to the maintenance of the ESL's watertight property. Once the integrity of the glycocalyx has been damaged, however, various cascades leading to permeability are activated [12, 13]. The digestion of the glycocalyx alone may be insufficient as a biological switch necessary for the opening of the tight junctions or the activation of other fluid transport systems. The cleavage of GAGs alone does not reportedly change the permeability of the pulmonary endothelium [14]. Thus, whether GAGs themselves play a role in the biological regulation of permeability remains uncertain.

To understand the mechanisms of permeability regulation in endothelial cells more thoroughly, the biological properties of the permeability control system must be further explored. We examined whether the digestion of GAGs alone could induce hyperpermeability using both *in vitro* and *in vivo* studies and compared findings obtained under both control and septic conditions. We also explored the genetic alterations that may control endothelial permeability through means other than the structural seal of the glycocalyx.

## Materials and methods

### *In vivo* study

All animal experiments were conducted with the approval of the Institutional Animal Care and Use Committee of Showa University (permit No. 18050). The experiments were performed using 6- to 8-week old male C57BL/6J wild-type mice (CLEA Japan, Inc., Tokyo, Japan) weighing 19 to 25 g. The mice were kept in a TPX cage (TM-TPX-10-VIII (8); Tokiwa Scientific Instrument Inc., Tokyo, Japan) under a 12-hour light-dark cycle with free access to water and standard chow (Labo MR Stock; Nosan Corporation, Kanagawa, Japan) under controlled temperature ($24°C \pm 1°C$) and humidity ($64\% \pm 1\%$) conditions. Mice were anesthetized via the intraperitoneal administration of a mixture of midazolam (4 mg/kg), medetomidine (0.75 mg/kg), and butorphanol (5 mg/kg). The animals were randomly apportioned into three groups: a control group, a septic model group, and a GAGs-digestion group.

For the GAGs-digestion mouse model, GAGs in the vascular endothelium of murine lung were degraded using a mixture of hyaluronidase (0.24 U/μL), heparinase III ($1.8 \times 10^{-2}$ U/μL), and neuraminidase ($2.07 \times 10^{-4}$ U/μL) in 200 μL phosphate buffered saline (PBS) administered via the tail veins of mice under anesthesia, according to a previously reported method [15]. One hour after the injection of the enzymes, the mice were used for the GAGs-digestion model [15].

For the septic mouse model, a previously reported lipopolysaccharide (LPS)-induced septic model was used [16–18]. In the present study, the mice were intraperitoneally injected with LPS from *Escherichia coli* O26:B6 (Sigma-Aldrich, St. Louis, MO, USA) dissolved in saline at a dose of 2 mg/kg at 0 and 18 h [18, 19]. Twenty-four hours after the first LPS injection, the mice were used for the septic model [18]. This regimen was confirmed to produce septic mice in our previous study [18].

**Observation of GAGs in mice.** Transmission electron microscopy (TEM) was used to examine the blood vessel endothelial cells in the lungs of mice in the control, GAGs-digestion model, and septic model groups. Lanthanum fixation was used because the lanthanum ion ($La^{3+}$) binds to GAGs [18]. Anesthetized mice were perfused via the heart with a fixative/staining solution (2% glutaraldehyde [Sigma-Aldrich, St. Louis, MO, USA], 0.1-M 4-(2-hydroxyethyl)-1-piperazineethanesulfonic acid buffer [Life Technologies, Carlsbad, CA], and 2% lanthanum (III) nitrate hexahydrate [Sigma-Aldrich, St. Louis, MO, USA]) after blood removal by perfusion with saline [18]. Diced pieces of the lung were fixed with 2% paraformaldehyde, 2% glutaraldehyde (Sigma), and 2% lanthanum (III) nitrate hydrate in 0.1-M sodium cacodylate buffer (pH7.4) and washed 3 times with 0.2-M sodium cacodylate. The diced pieces were post-fixed with 1% osmium tetroxide for 60 min, dehydrated through an ethanol series, and then embedded in epoxy resin. Ultrathin sections (70 nm thick) were then stained with uranyl acetate and lead citrate and observed under a transmission electron microscope (H-7600; Hitachi); 3 images were quantified per mouse. The thickness of lanthanum staining was measured at 10 locations per image in each group by transferring the TEM images to analysis software (ImageJ; NIH, Bethesda, MD), and the mean thickness (nanometers) of the stained area was defined as the GAGs thickness [18].

**Determination of vascular permeability.** FITC-labeled dextran (2,000 kDa; Sigma-Aldrich, St. Louis, MO, USA) dissolved in saline was administered via the tail vein under anesthesia to each mouse (n = 7 for control; n = 7 for septic model; n = 7 for GAGs-digestion model) to visualize the leaked high molecular substance [20]. The lungs were removed 15 minutes after dextran intravenous administration, then immediately embedded in Tissue-Tek O.C.T. compound (Sakura Finetek, Tokyo, Japan) and frozen. Tissue sections of the lung were observed using a BZ-9000 fluorescence microscope (Keyence). The camera settings (magnification: 40) and exposure times (fluorescence images of FITC-labeled dextran: 1/8; bright field images: 1/1000) were the same for all the images. The area of the lung and vessels were randomly analyzed. The fluorescence intensity of FITC-labeled dextran in nonvascular tissue was measured for each group by transferring the images to analysis software (ImageJ; NIH, Bethesda, MD). First, the area of extravascular fluorescence was measured using a representative septic model. The obtained area was applied to the other images. The averaged value of the intensity was presented for each group.

## *In vitro* study

To measure endothelial permeability quantitatively, we also performed an *in vitro* study, since *in vitro* models are more sensitive at detecting vessel wall permeability than *in vivo* models.

Human umbilical vein endothelial cells (HUVECs; Promocell, Heidelberg, Germany) were seeded in cell culture flasks containing low-serum (2% v/v) endothelial growth medium (Promocell, Heidelberg, Germany) and an antibiotic and antimycotic solution (penicillin, streptomycin and amphotericin B; Sigma-Aldrich, St. Louis, MO, USA) [15]. In all the experiments, HUVECs from the same origin were used after 4 to 6 passages [15].

To analyze the formation of GAGs, HUVECs were cultured in 4-well LabTek slides ($1.3 \times 10^4$ cells/well; Nunc, Rochester, NY, USA) for 10 days using a previously reported

method [15]. The HUVECs were treated for 1 h in medium containing hyaluronidase from bovine testes (25 U/mL; Sigma-Aldrich, St. Louis, MO, USA), heparinase III from *Flavobacterium heparinum* (2 U/mL; Sigma-Aldrich, St. Louis, MO, USA), and neuraminidase from *Clostridium perfringens* (0.83 U/mL; Sigma-Aldrich, St. Louis, MO, USA) [15]. The HUVECs were then fixed in 2% paraformaldehyde and stained with fluorescein isothiocyanate (FITC)-labeled wheat germ agglutinin (WGA)-lectin (1 mg/mL; Sigma-Aldrich, St. Louis, MO, USA). The fluorescence images were acquired using a BZ-9000 fluorescence microscope (Keyence). The same camera settings and exposure times were used for all the images.

HUVECs were seeded in 12-well Transwell Permeable Supports (12-mm insert, 12-well plate, 0.4-μm polyester membrane; Costar; Corning, NY, USA) at a density of $5 \times 10^4$ cells/well and cultured for 10 days. The transendothelial electrical resistance (TEER) was measured in the medium at 37°C using tissue resistance measurement chambers for tissue culture cups (Endohm-12 chamber; World Precision Instruments) and a resistance meter (EVOM2; World Precision Instruments) [21]. The TEER was calculated based on Ohm's law and was expressed as a percentage of the basal level [21]. The basal TEER of the HUVEC monolayers was 19 to 25 $\Omega$/cm$^2$ [21]. Histamine ($3 \times 10^{-6}$ mol/L; Sigma-Aldrich, St. Louis, MO, USA) was used as a positive control [21]. The effect of GAGs digestion on the TEER was evaluated.

## Gene expression profiles (*in vivo* study)

To explore the biological mechanism responsible for the regulation of permeability, we also examined the genetic expression of endothelial cells. Lung tissues from each group of mice were excised (n = 3 for control; n = 3 for septic model; n = 3 for GAGs-digestion model). The lungs were then minced with scissors and placed in 5 mL of PBS containing 0.5% fetal bovine serum (FBS) and 5-mM EDTA in 15-mL tubes; mechanical distribution was then performed using the gentle MACs Dissociator (Miltenyi Biotec) for 3 min and 28 s before filtration and passaging through a 100-μm cell strainer. The samples were then centrifuged and resuspended three times for subsequent staining. For the surface staining, the cell suspensions were incubated with an allophycocyanin (APC) anti-mouse CD31/platelet and endothelial cell adhesion molecule-1(Pecam-1) antibody with isotype control (1:100, Milteny Biotec) and a phycoerythrin (PE) anti-mouse hematopoietic lineage cocktail with isotype control (1:50, BioLegend) in PBS with 0.5% FBS for 20 min. Finally, the stained cells were suspended in PBS containing 0.5% FBS and propidium iodide (PI, 1 μg/mL), and the cells were sorted using a FACSAria II (BD Biosciences). All the gates were designed based on the isotypes and single-stain controls.

For the quantitative real-time polymerase chain reaction, RNA from the sorted cells was isolated using the RNeasy® Plus Mini Kit (Qiagen). Reverse transcription was performed using SSC VILO (Invitrogen), and quantitative real-time polymerase chain reaction (qPCR) was performed using the cDNA samples and a 7500 detection system (n = 3 for CD31-positive sorted cells in control mice; n = 3 for CD31-negative sorted cells in control mice; Invitrogen). The quantification of the samples was calculated according to the threshold cycle using the ΔΔCt method. The CD31 primer design was as follows: forward primer, 5′-CTGCCAGTCC GAAAATGGAAC-3′; reverse primer, 5′-CTTCATCCACTGGGGCTATC-3′.

**RNA sequencing.** A library for RNA sequencing was prepared using the SMARTer® Stranded Total RNA-sequence (RNA-Seq) Kit v2—Pico Input Mammalian (Z4411N; TaKaRa), according to the manufacturer's instructions. Paired end sequencing (read1: 50 nt and read2: 25 nt) was performed using the Illumina NextSeq 500 system. STAR [22]. v.2.6.1a was used to map the filtered reads to the GRCm38.p6 comprehensive annotation. The reads on annotated genes were counted using featureCounts [23]. Genes with differential expressions were estimated using DESeq2 [24], v.1.30.1, and were selected using an adjusted *P* value

of <0.05 and a log2 fold change of ≧2.0. A gene ontology (GO) enrichment analysis of the genes with differential expressions was performed using DAVID, v.6.8.

**Statistical analyses.**   The measured data were presented as the mean ± standard deviation. Comparisons were made using an unpaired Student *t*-test for the fluorescence intensity of FITC-labeled WGA-lectin for HUVECs and qPCR in endothelial cells. TEER data for the HUVECs, the fluorescence intensity of FITC-labeled dextran in nonvascular lung tissue, and the GAGs thickness analysis of TEM images were evaluated using a one-way analysis of variance (ANOVA), followed by the Tukey honestly significant difference (HSD) test. Probability values of less than 0.05 were considered significant. The statistical analyses were conducted using Microsoft Excel for office 365 MSO.

## Results

### GAGs in lung vascular endothelial cells were digested by enzymatic treatment

TEM images of the lung vessels in normal control mice revealed a lanthanum-positive GAGs layer, while most of the GAGs layers had been digested from the endothelium in the enzyme-injected mice as well as in the LPS-administered septic mice (Fig 1A–1C). Calculations of the GAGs thickness using ImageJ software showed that the mean thickness was significantly lower in the enzyme-injected mice and the LPS-administered septic mice than in the normal control mice, and no significant difference in the mean thickness was seen between the enzyme-injected mice and the LPS-administered septic mice (Fig 1D). These results suggested that enzymatic digestion using a mixture of neuraminidase, heparinase III, and hyaluronidase was useful for degrading the glycocalyx on endothelial cells in a mouse model.

### Degradation of the endothelial glycocalyx did not affect the vascular permeability of the pulmonary blood vessels

To examine whether degradation of the endothelial glycocalyx alone increases vascular hyperpermeability, FITC-labeled dextran was administered via the tail vein in mice and the exudation of FITC-labeled dextran from lung venules was visualized. No exudation of FITC-labeled dextran was observed in the control mice and enzyme-injected mice, while the exudation of FITC-labeled dextran was apparent in LPS-administered septic mice (Fig 2A–2C, n = 7 for control; n = 7 for GAGs-digestion model; n = 7 for septic model). Calculations of the fluorescence intensity using ImageJ software showed no significant difference in the mean intensity among the enzyme-injected and control mice, whereas a significant increase was observed in the LPS-injected septic mice (Fig 2D, *P* = 5.257e-05, compared with the septic model, n = 7 for control; n = 7 for GAGs-digestion model; n = 7 for septic model). These results suggest that the vascular hyperpermeability observed in the septic mouse model did not occur in the GAGs-digestion mice.

### Enzymatic degradation of the glycocalyx had no effect on the permeability of HUVECs

To examine whether the degradation of the glycocalyx affects vascular permeability *in vitro*, a HUVECs model was used because *in vitro* studies using electron microscopy and fluorescence microscopy have shown that HUVECs possess a glycocalyx [15, 25]. In the present study, GAGs constituting the glycocalyx were enzymatically digested. In the fluorescence analysis using FITC-labeled WGA-lectin, the digested HUVECs had a much lower FITC-fluorescence than those without digestion (Fig 3A). In addition, the fluorescence intensity was calculated

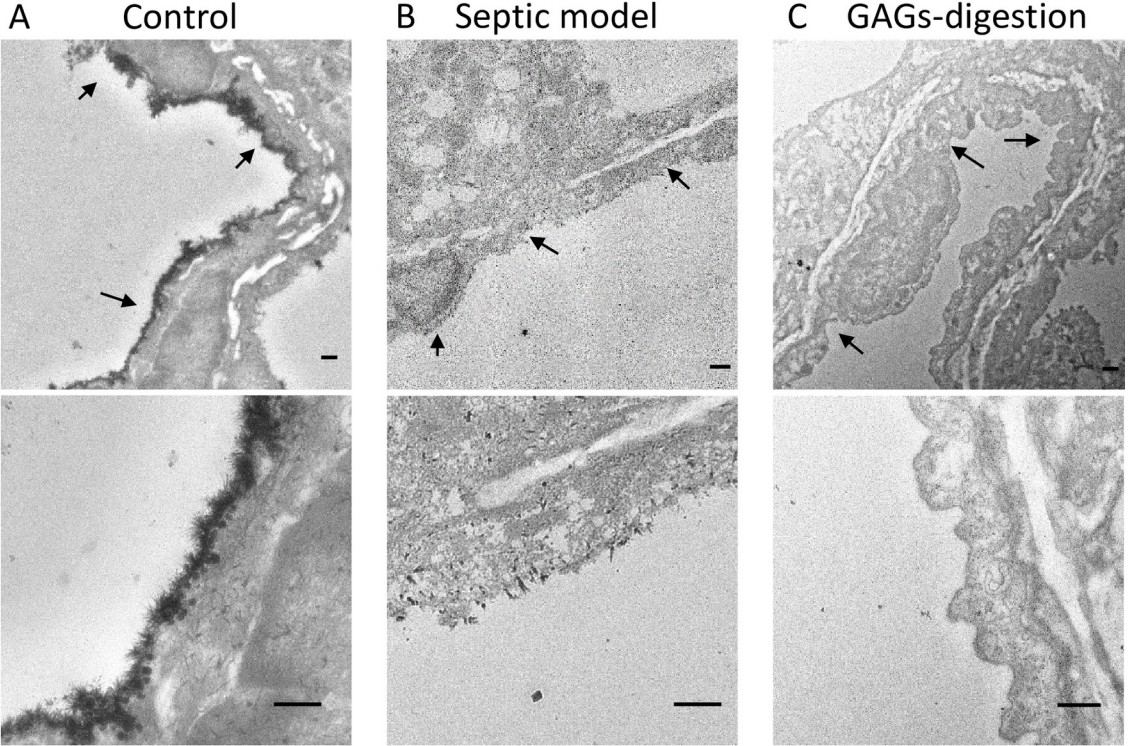

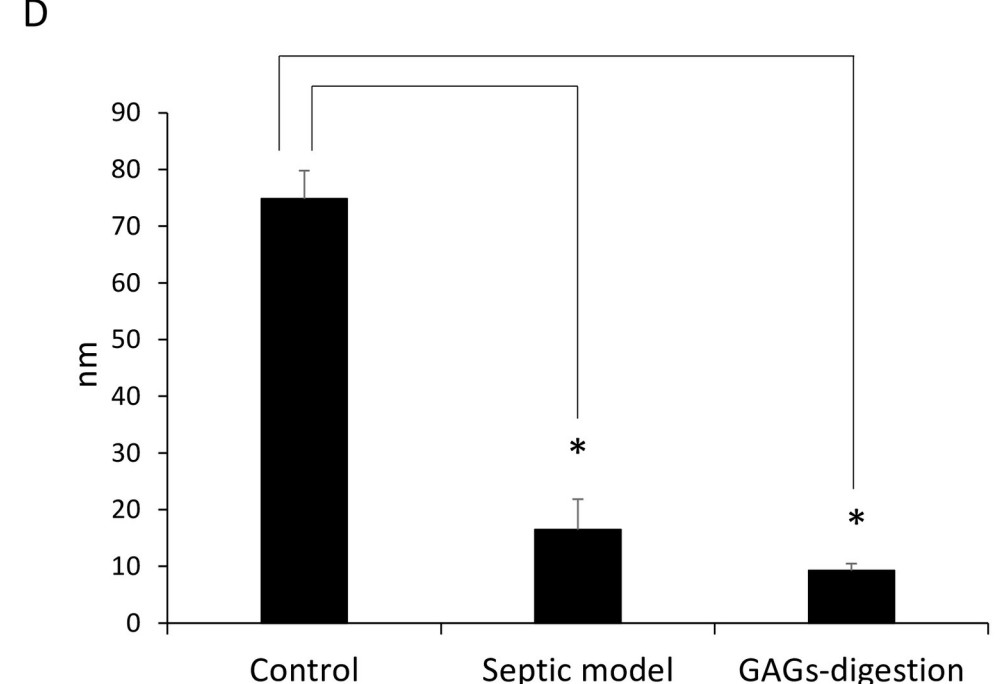

**Fig 1. GAGs visualized using transmission electron microscopy (TEM).** (A) Low magnification (upper) and high magnification (lower) TEM images of the lung vessels in control mice are shown. The arrows indicate lanthanum-positive GAGs in the vascular lumens. The scale bars in each image are 500 nm (n = 3 for control). (B) GAGs were absent in the septic mouse model, compared with the control mice. The arrows indicate the vascular lumens (n = 3 for septic model). (C) GAGs were absent in the GAGs-

digestion mice, compared with the control mice. The arrows indicate the vascular lumens (n = 3 for GAGs-digestion model). (D) Mean thickness of GAGs. The thickness was significantly less in the septic mice and the GAGs-digestion mice than in the control mice ($^*P < 0.05$ compared with control, one-way ANOVA followed by Tukey's HSD, n = 3 for control; n = 3 for septic model; n = 3 for GAGs-digestion model).

using ImageJ software. The mean intensity of the GAGs-digested HUVECs was also significantly lower than that of the control HUVECs (Fig 3B, $P = 0.029212854$, n = 3 for control; n = 3 for GAGs digestion). These data suggested that enzymatic digestion of the HUVECs caused glycocalyx degradation. Next, to evaluate whether the effect of glycocalyx digestion changed the permeability of the endothelial cells, the TEER of confluently cultured HUVECs was measured. No significant difference in the TEER was seen between HUVECs with and those without digestion (Fig 3C, n = 3 for control; n = 3 for GAGs digestion). The HUVECs with or without GAGs digestion exhibited a rapid 10% decrease in the TEER after 2 minutes the addition of histamine ($3 \times 10^{-6}$ mol/L), compared with untreated HUVECs (Fig 3C, $P = 0.0007866$, n = 3 for control; n = 3 for control+Histamine; n = 3 for GAGs digestion+Histamine), while no significant difference in the TEER after the addition of histamine was seen between HUVECs with and those without digestion (n = 3 for control+Histamine; n = 3 for GAGs digestion+Histamine). A previous study reported that the TEER after histamine stimulation recovered a few minutes after stimulation [21]. Here, the TEER after histamine administration recovered 4 minutes after the addition of a solution with or without GAGs digestion. These data suggested that GAGs digestion degraded the glycocalyx but did not affect the permeability of endothelial cells *in vitro*.

## Comparison of gene expression profiles in endothelial cells

The endothelial cells were purified from the lungs of control, enzyme-injected, and LPS-injected mice using an APC anti-CD31 antibody and flow cytometry (Fig 4A). LPS-injected mice were used as a septic mouse model [18, 19]. In this study, the gene changes in the vascular endothelial cells induced by GAGs digestion were compared with those in a control and a septic model. qPCR confirmed that CD31-positive sorted cells showed significantly higher expression levels of the CD31 gene, compared with CD31-negative cells (Fig 4B). RNA-seq was used to generate the gene expression profiles of each of the sorted cells (S1 Table). A hierarchical cluster analysis demonstrated that the gene expression profiles of the GAGs-digested endothelial cells were different from the control mice, although they were more similar to those in control mice than to those in septic mice (Fig 4C). In addition, a Venn diagram showing direct comparisons between GAGs-digested endothelial cells and control endothelial cells indicated that the expressed genes (FPKM>5) were similar in both cell types (Fig 4D). The gene ontology (GO) analysis showed that inflammation-related gene ontologies including the positive regulation of the inflammatory response and the cytokine-mediated signaling pathway were significantly enriched in the upregulated genes of GAGs-digested endothelial cells (Table 1). However, genes that were directly associated with endothelial leakage, such as CDH5 (also known as VE-cadherin) and CD31, were not included among the differentially expressed genes when the GAGs-digested endothelial cells and the control endothelial cells were compared (S1 Table). These data suggested that GAGs-digestion stimulated the proinflammatory response of endothelial cells without inducing vascular hyperpermeability.

## Discussion

The endothelial glycocalyx layer acts as a barrier against the leakage of fluid and large molecules [26]. The endothelial glycocalyx plays a crucial role in preventing edema formation and sustaining the microcirculatory environment through a mechano-sensing system [27].

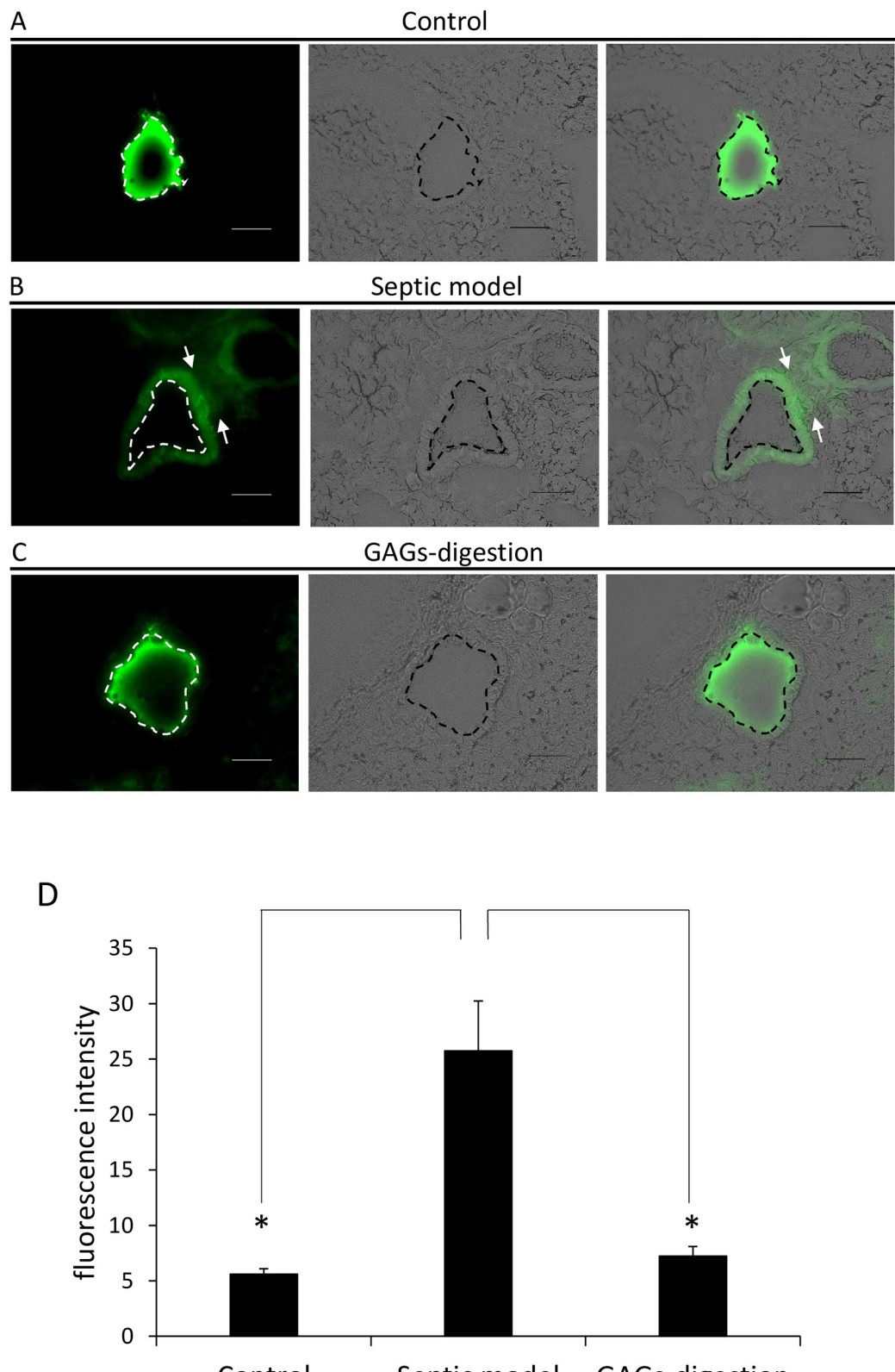

**Fig 2. Vascular permeability in the lung after lipopolysaccharide (LPS) administration or enzyme administration.**
(A) Fluorescence images of lung tissue after FITC-labeled dextran administration via the tail vein in control mice. A

fluorescence image of FITC-labeled dextran in lung vessels is shown on the left. A bright field image is shown in the middle. A merged image of the fluorescence image of FITC-labeled dextran and the bright field image is shown on the right. The control mice did not show evidence of the leakage of FITC-labeled dextran (n = 7 for control). (B) Fluorescence images of lung tissue after FITC-labeled dextran administration via the tail vein in a septic mouse model. The septic model mice exhibited the leakage of FITC-labeled dextran in nonvascular tissue, compared with the mice in the other groups. The arrows indicate the FITC-labeled dextran in the nonvascular tissue (n = 7 for septic model). (C) Fluorescence images of lung tissue after FITC-labeled dextran administration via the tail vein of GAGs-digestion mice. GAGs-digestion mice did not exhibit the leakage of FITC-labeled dextran (n = 7 for GAGs-digestion model). (D) Fluorescence intensity of FITC-labeled dextran in nonvascular tissue. The fluorescence intensity of FITC-labeled dextran in nonvascular tissue in septic model mice was significantly higher than that in the other groups ($^*P < 0.05$, compared with the septic model, one-way ANOVA followed by Tukey's HSD, n = 7 for control; n = 7 for septic model; n = 7 for GAGs-digestion model). The dotted lines indicate the vascular lumens. The scale bars are 50 μm.

Reportedly, the pulmonary endothelial surface layer is significantly thicker than that in other organs, suggesting a unique physiological function [14]. However, the lung is a vulnerable organ and can become a target of sepsis even with such a sophisticated endothelial barrier [28]. The pathophysiology of sepsis involves leukocyte activation, which may play a significant role in barrier disruption, and the disruption of the glycocalyx may be associated with lung edema formation. Therefore, whether the glycocalyx per se is just a structural barrier layer of the endothelium or plays a functional role in vascular permeability should be elucidated. In the present study, we demonstrated that the enzymatic digestion of GAGs alone was insufficient to increase the vascular permeability, and the impact of GAGs digestion on permeability differed largely from that occurring in a septic model.

We used an enzyme challenge for GAGs digestion. Previous studies have demonstrated that HUVECs have a glycocalyx and that enzymatic treatment leads to the digestion of GAGs [15, 25]. Although it has been reported that GCX thickness differs substantially between ex vivo and in vitro [25], the existence of the GCX has been confirmed. To evaluate permeability quantitatively, we used HUVECS in the present study. The endothelial glycocalyx consists of glycoproteins and PGs with GAGs side chains. The most prominent GAG on the surface of the endothelial cells is heparan sulfate, with the remainder comprised of chondroitin sulfate and hyaluronic acid. In this study, we used an enzymatic mixture consisting of hyaluronidase, heparanase, and neuraminidase to degrade the GAGs effectively. We confirmed GAGs digestion using a fluoroscopic examination *in vitro* and an ultrastructural examination *in vivo*. However, we failed to observe hyperpermeability even in these models.

Histamine is an important chemical mediator and neurotransmitter that induces hyperpermeability through the activation of the H1 receptor that regulates the overall response [29, 30]. Our study demonstrated that GAGs digestion in HUVECs did not have synergistic effects on hyperpermeability, which was originally driven by the addition of histamine. Thus, it should be noted that the H1 receptor is not protected by the glycocalyx. A previous study reported that the cleavage of heparan sulphate residues from the glycocalyx in fenestrated glomerular endothelial cells did not affect the TEER [31]. These previous results agree with the presently reported findings. Thus, another biological barrier, rather than a structural barrier such as the glycocalyx, may play a key role in the onset of hyperpermeability. Neutrophil rolling and adhesion to the endothelial surface have been reported to precede hyperpermeability [5]. Neutrophils migrate though the cleft of the pulmonary endothelium into the alveolus during pulmonary edema. Conceivably, neutrophils might play a key role in pulmonary vascular permeability. However, the biological mechanism regulating the hyperpermeability of the pulmonary endothelium has not yet been elucidated. Moreover, GAGs digestion itself reportedly plays a biological role in permeability [32]. Therefore, the digestion of GAGs may help to transduce signals into the endothelium, increasing permeability.

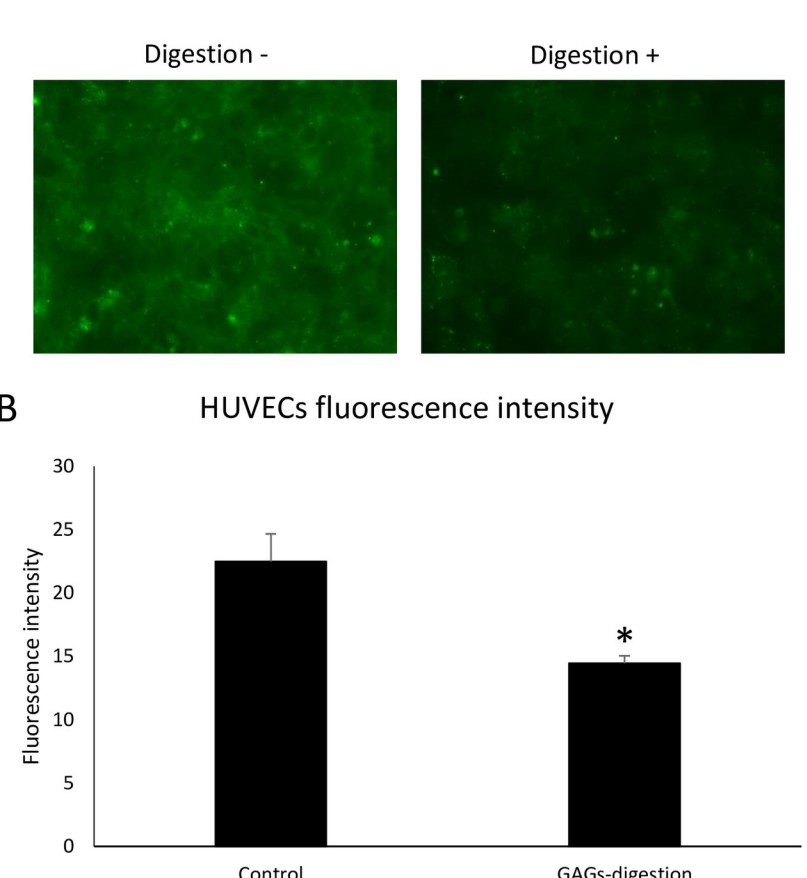

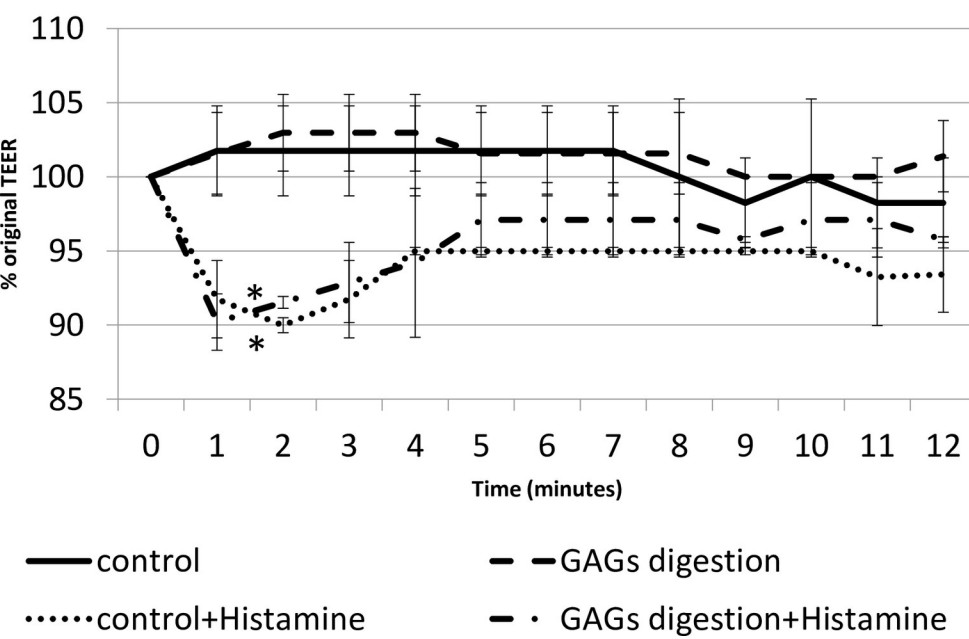

**Fig 3. Effect of enzymatic degradation and histamine on the permeability of HUVECs.** (A) Human umbilical vein endothelial cells (HUVECs) with or without glycosaminoglycans (GAGs) digestion. HUVECs with GAGs digestion

(right, n = 3 for GAGs digestion) exhibited a lower fluorescence intensity of fluorescein isothiocyanate (FITC)-labeled wheat germ agglutinin (WGA)-lectin than control HUVECs (left, n = 3 for control). (B) Fluorescence intensity of FITC-labeled WGA-lectin. HUVECs with GAGs digestion exhibited a significantly lower fluorescence intensity than the control cells (*$P < 0.05$, unpaired t-test, n = 3 for control; n = 3 for GAGs digestion). (C) The transendothelial electrical resistance (TEER) was measured in real-time in confluent monolayers of HUVECs. The addition of histamine to the HUVECs reduced the mean TEER by 10% relative to that in the control, while GAGs digestion had no significant effect on the TEER of HUVECs (*$P < 0.05$ compared with control, one-way ANOVA followed by the Tukey HSD, n = 3 for control; n = 3 for GAGs digestion; n = 3 for control+Histamine; n = 3 for GAGs digestion +Histamine).

Previous studies have shown that inflammation increases vascular permeability, allowing the passage of plasma contents up to 2000 kDa in size [20], and HUVECs exhibited a rapid decrease in the TEER after the addition of LPS [21]. In our results, the LPS-induced septic condition caused GAGs digestion and macromolecular hyperpermeability, whereas GAGs digestion induced by enzymatic treatment did not affect the macromolecular permeability. GAGs digestion induced the expressions of several proinflammatory genes including Ppbp, S100A9, Elane, Adam8, and Itgam. Consistent with this finding, glycocalyx degradation reportedly induces a proinflammatory phenotype [33]. A previous study demonstrated that the loss of the glycocalyx as a result of inflammation exposes previously hidden endothelial surface adhesion molecules, including ICAM-1 and VCAM-1, leading to the neutrophil recognition of and adhesion to the endothelial surface [5]. Therefore, GAGs digestion might be involved in the initial step of the inflammatory response in endothelial cells. Further studies are required to distinguish direct effects due to the loss of the glycocalyx from the effects due to cell-cell interactions.

A previous study reported that glycocalyx digestion by heparinase III did not affect TEER of a glomerular endothelial cell line, while increased the albumin passage across the monolayers [31]. A particular issue is that dextrans as large as 2000K may not be useful probe molecules for more subtle, but still clinically important, changes in vascular permeability and measurements of TEER are insensitive to changes in glycocalyx permeability because the glycocalyx does not significantly restrict the small ion diffusion upon which this method is based. Therefore, this finding is partly consistent with our result despite using different cell lines. However, we did not examine the albumin passage. Therefore, further studies will be needed to clarify this in the future. In our in vivo model, we only explored vascular permeability in the lung. We chose the lung because pulmonary vessels are particularly susceptible to septic conditions. However, the extrapolation of our results to vessels in other organs requires caution.

Recently, the concept of fluid therapy has gradually changed. Chappell et al. proposed a rational approach to fluid therapy, introducing type 1 and type 2 fluid leakage (filtration) [34]. The former concept of fluid therapy recommended the use of supplemental fluid to avoid a fluid deficit. However, the simple infusion of a crystalloid has gradually been recognized as not contributing to an expansion of the blood volume; thus, fluid supplementation is not an efficient way to optimize the intravascular blood volume [35]. The mechanisms of type 1 and type 2 fluid filtration have now begun to attract attention, rather than the concept of how to supply fluid. Fluid absorption, representing an expansion of the intravascular volme, cannot be expected to occur even when the infusion volume increases. The expansion volume is difficult to predict because most intravenously administered fluid is filtrated. Thus, the scope for updating fluid therapy should focus on the mechanism by which this fluid leaks and how such leakage should be counteracted. Our study could provide information pertinent to the prevention of intravascular volume loss. Elucidating the molecular mechanism of vascular permeability could contribute to the development of rational fluid therapy, since pathological vascular leakage is the main pathological entity that leads to difficulties in fluid management.

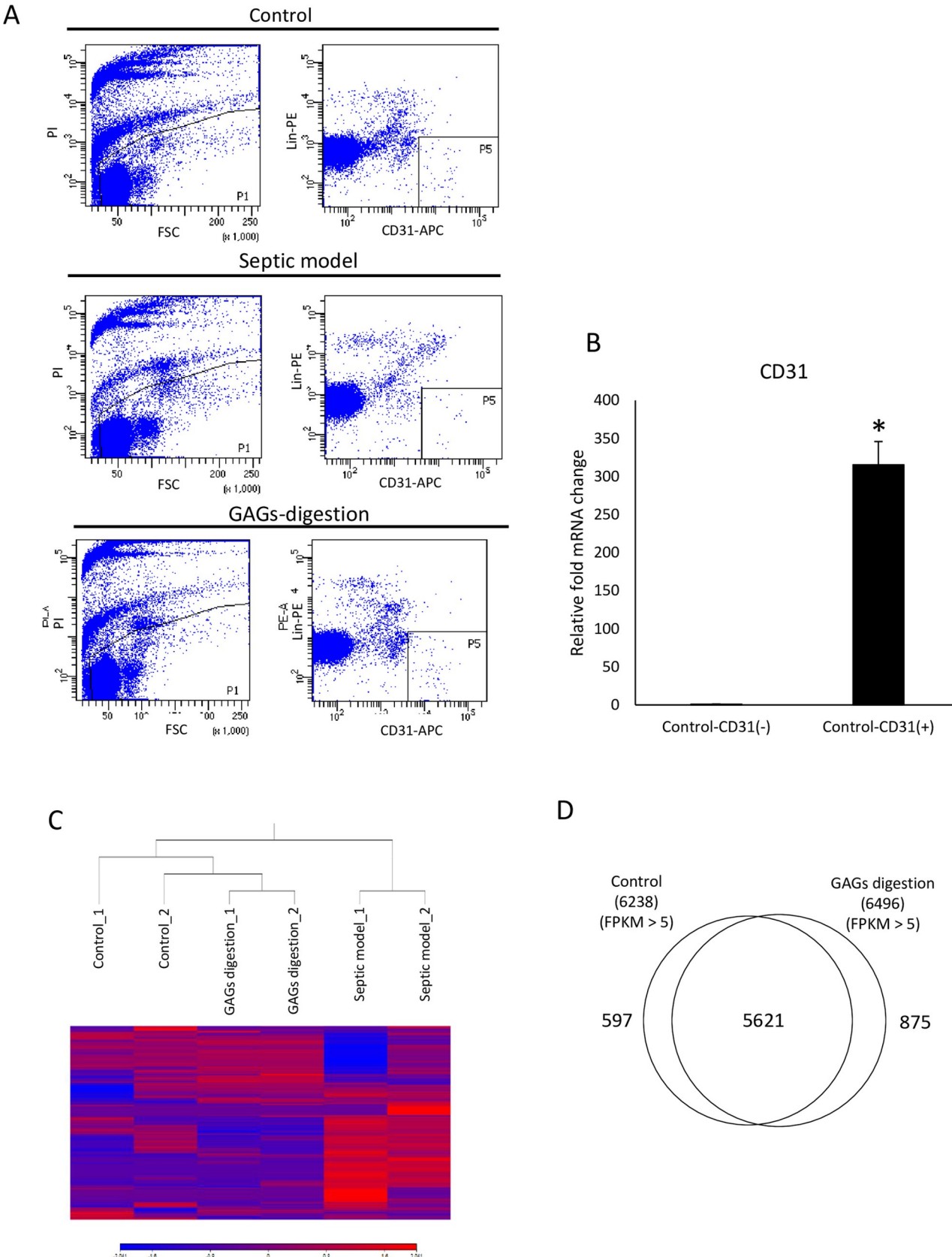

**Fig 4. Gene expression profiles of endothelial cells from control and GAGs-digestion mice.** (A) Representative plots demonstrating the gate hierarchy for the isolation of endothelial cells from dissociated lung tissue. Dead cells, doublets, and aggregates were gated out by forward scatter, side scatter, and propidium iodide (PI) staining, respectively. Gating for PI-negative live cells is shown on the left, and CD31-positive and lineage markers-negative cells were gated as endothelial cells, as shown on the right. (B) Quantitative real-time polymerase chain reaction for CD31. The gene expression of CD31 was significantly higher in the CD31-positive sorted cells than in the CD31-negative sorted cells (*$P < 0.05$, unpaired t-test, n = 3 for CD31-positive sorted cells in control; n = 3 for CD31-negative sorted cells in control). (C) Hierarchical cluster analysis based on the global gene expression examined through RNA sequencing. The data are shown for the sorted endothelial cells of the control group (n = 2), the septic model group (n = 2), and the GAGs-digestion group (n = 2). (D) Venn diagram illustrating the expressed genes (FPKM>5) detected in the control and the GAGs-digestion mice.

**Table 1. Gene ontology (GO) terms enriched in differentially expressed genes upregulated in glycosaminoglycans (GAGs)-digested endothelial cells relative to control endothelial cells.**

| GO term | *P* Value | Genes |
|---|---|---|
| Leukocyte migration involved in inflammatory response | 3.40E-05 | PPBP, S100A9, ELANE, ITGAM |
| Neutrophil chemotaxis | 4.04E-04 | CCL22, PPBP, S100A9, CCL9, ITGAM |
| Lipid metabolic process | 5.03E-04 | CHKA, ALOX15, LIPO1, APOE, SORL1, PLA2G7, PLBD1, PLPP1, PLCXD2, SMPD2 |
| Inflammatory response | 0.001750203 | SLC11A1, CCL22, MEFV, LTB4R1, S100A9, PLA2G7, CCL9, IGFBP4 |
| Positive regulation of inflammatory response | 0.003937251 | ACE, PDE2A, S100A9, CCL9 |
| Cytokine-mediated signaling pathway | 0.006334371 | EREG, IL20RB, SOCS3, CX3CR1, JAK3 |
| Chemotaxis | 0.021722293 | CCL22, S100A9, CX3CR1, CCL9 |
| Regulation of smooth muscle cell migration | 0.029822476 | ACE, SORL1 |
| Chemokine-mediated signaling pathway | 0.031305524 | CCL22, PPBP, CCL9 |
| Negative regulation of neuron death | 0.032360588 | APOE, SORL1, TIGAR |
| Positive regulation by host of viral process | 0.03470664 | APOE, CEACAM1 |
| Negative regulation of beta-amyloid formation | 0.03470664 | APOE, SORL1 |
| Ammonium transport | 0.03470664 | SLC12A2, AQP1 |
| Positive regulation of interleukin-8 biosynthetic process | 0.039566485 | PRG3, ELANE |
| cGMP-mediated signaling | 0.039566485 | PDE2A, APOE |
| Low-density lipoprotein particle remodeling | 0.04440213 | APOE, PLA2G7 |
| Positive regulation of protein binding | 0.046058172 | ACE, TIAM1, HFE |

In conclusion, we have demonstrated that GAGs digestion itself may be insufficient to induce hyperpermeability, suggesting that further steps are required to increase endothelial permeability. Although we failed to confirm the mechanism responsible for regulating the permeability of the glycocalyx, GAGs digestion enabled us to identify the footprints of several signal transductions in endothelial cells. Taken together, our results suggest that the glycocalyx constitutes a structural barrier affecting fluid permeability and may play an important role in initiating signal transductions. Further study is required to explore whether the resulting signaling is related to the opening of tight junctions, leading to hyperpermeability.

## Supporting information

**S1 Checklist. ARRIVE guidelines checklist.**
(PDF)

**S1 Table. Gene expression profiling of each of the sorted cells using RNA-seq.**
(DOCX)

## Author Contributions

**Conceptualization:** Kenji Mishima, Takehiko Iijima.

**Data curation:** Kyoko Abe, Junichi Tanaka.

**Formal analysis:** Kyoko Abe, Junichi Tanaka.

**Methodology:** Kyoko Abe, Junichi Tanaka.

**Project administration:** Kyoko Abe, Junichi Tanaka, Kenji Mishima, Takehiko Iijima.

**Writing – original draft:** Kyoko Abe, Junichi Tanaka, Kenji Mishima, Takehiko Iijima.

**Writing – review & editing:** Junichi Tanaka, Kenji Mishima, Takehiko Iijima.

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
