## [Decision Letter · Decision Letter 0]

29 Jan 2021

PONE-D-20-40689

Exploring the mechanism of hyperpermeability following glycocalyx degradation: Beyond the　glycocalyx as a structural barrier

PLOS ONE

Dear Dr. Iijima,

Thank you for submitting your manuscript to PLOS ONE. After careful consideration, we feel that it has merit but does not fully meet PLOS ONE’s publication criteria as it currently stands. Therefore, we invite you to submit a revised version of the manuscript that addresses all the points raised during the review process.

Both reviewers agree, that the topic, the role of glycocalyx in endothelial permeability is interesting, but the study needs further experiments and additional controls to clarify the exact role of junctional opening, glycocalyx loss or cellular interactions in the present observations. The conclusion need to be amended, too.

We look forward to receiving your revised manuscript.

Kind regards,

Mária A. Deli, M.D., Ph.D.

Academic Editor

PLOS ONE

Journal Requirements:

2. We noticed you have some minor occurrence of overlapping text with the following previous publications, which needs to be addressed:

- https://www.jstage.jst.go.jp/article/sujms/31/2/31_167/_article

- https://bmcmolbiol.biomedcentral.com/articles/10.1186/s12867-015-0029-5

In your revision ensure you cite all your sources (including your own works), and quote or rephrase any duplicated text outside the methods section.

Further consideration is dependent on these concerns being addressed.

3. To comply with PLOS ONE submissions requirements, in your Methods section, please provide additional information on the animal research and ensure you have included details on (i) methods of sacrifice, (ii) methods of anesthesia and/or analgesia, and (3) efforts to alleviate suffering.

'This study was supported in part by Grants-in-Aid for Scientific Research [Grant 16K11762] from the Ministry of Education, Culture, Sports, Science and Technology of Japan.'

'The authors received no specific funding for this work.'

Reviewers' comments:

Reviewer's Responses to Questions

**Comments to the Author**

1. Is the manuscript technically sound, and do the data support the conclusions?

Reviewer #1: Partly

Reviewer #2: No

2. Has the statistical analysis been performed appropriately and rigorously? 

Reviewer #1: Yes

Reviewer #2: N/A

3. Have the authors made all data underlying the findings in their manuscript fully available?

Reviewer #1: Yes

Reviewer #2: Yes

4. Is the manuscript presented in an intelligible fashion and written in standard English?

Reviewer #1: Yes

Reviewer #2: Yes

5. Review Comments to the Author

Reviewer #1: The authors exploring the glycocalyx degradation influence on the hyperpermeability of pulmonary vessel cells. However, no clear conclusion was made tightly related to the analysis. More statistics analysis and a decent conclusion is needed.

Minor comments:

1. Page 6, line 66, "One hour after the injection of the enzymes,..." and line 71, "Twenty-four hours after the first LPS injection,...". An proof of time chosen here is needed, it can be a citation from previous study or a time dependence test.

2. Page 13, Fig. 1 needs to indicate how many instances are included in this figure.

3. Page 15, line 217. "The arrows. indicate the FITC-labeled dextran...", there is no arrow in Fig. 2.

Major comments:

1. Page 14, "Furthermore, FITC-labeled dextran was not exuded from the venules in enzyme-injected mice, ... LPS-injected septic mice (Fig. 2D)." In this section, firstly no difference was mentioned between Fig. 2B and C, then another statement of no significant difference between Fig. 2A and C was made. It is confusing. A clear statement of comparing among Fig. 2 A-C should be made. In addition to this, there is no statement indicates whether Fig. 2A-C is based only on one instance or averaged result. If it is based on only one replica, a statistically analysis is needed.

2. Page 15, "The mean intensity of the GAGs-digested HUVECs was also significantly ... endothelial cells in vitro." In this section, "significant" was used to show the difference in Fig. 3B and "no significant difference" to show Fig. 3C. A more clear definition is needed for how much can be called "significant". A quantitive result is helpful. Moreover, in Fig. 3C, as time goes, the differences become larger (12 mins) than short time result (3 mins). A time dependent result change should be made for a solid conclusion. And an underline relationship explanation among panels in Fig. 3 is needed.

3. Page 23, "Our negative results may have arisen from the use of a different cell line." Either a prof of chosen of current cell line or a quantity analysis of result change based on varying cell lines is needed.

Reviewer #2: These investigations attempt to evaluate the changes in vascular permeability and gene expression in endothelial barriers in mouse lung and cultured HUVECs when the glycocalyx is removed under conditions of sepsis and enzyme degradation and histamine. Unfortunately the experimental conditions are not adequate to effectively examine these questions. In the permeability experiments the authors do not take into account that the glycocalyx and the endothelial cells lie in series so the properties of both barriers must be taken into account when interpreting the results. Thus the use of the very large dextran (MW 2000KD) is restricted by the endothelial barrier with intact junction which is likely to be the case when only enzyme degradation is investigated. The observation that there is no increased leakage of the dextran when the glycocalyx is removed tells nothing about the contribution of the glycocalyx to the normal barrier. On the other hand the sepsis conditions modify both glycocalyx and the endothelial integrity resulting in breakdown of both barriers.

The use of TEER are also a problem. The glycocalyx offers very little resistance to small ions. Most of the small reported resistance is due to the reduced area for transport through the inter-endothelial junctions. Thus the failure to see changes in TEER with enzyme degradation is the expected result and provides no information about the status of the junctions On the other hand the fall in resistance after histamine is exactly as expected when histamine transiently opens the junctions between endothelial cells.

Thus the permeability investigating need to be replaced by studies that use a lower molecular weight dextran to investigate large molecule permeability in both in vivo and invitro experiments.

The gene expression are potentially more interesting but again the experimental design limits the usefulness of the result from enzyme degradation. Specifically expression in mouse lung of enzyme digested glycocalyx was examined after only 1 hour after the digestion. The cells from the septic model had been exposed to inflammatory conditions for 24 hours. The observation that these there was no change in genes associated with endothelial leakage ( VE cadherin, PECAM1) were not differentially expressed over this period provides cannot be used to conclude that removal of the glycocalyx has no effect on gene regulating permeability. However the observation that genes associated with the cytokine mediated response are upregulated is of interest . However it is not clear where this is a direct result of glycocalyx removal, or activation of a range of other vascular cells which themselves interact with endothelial when their own glycocalyx is modified. Further control studies are required to distinguish some of these complex interactions.

6. PLOS authors have the option to publish the peer review history of their article (what does this mean?). If published, this will include your full peer review and any attached files.

Reviewer #1: **Yes: **Chuqiao Dong

Reviewer #2: **Yes: **FitzRoy E Curry PhD

---

## [Author Response · Author response to Decision Letter 0]

28 Apr 2021

This manuscript has been resubmitted (the manuscript number was PONE-D-20-40689). 

Thank you for taking the time to review our work and provide constructive feedback. We attach a version showing the tracked changes and, separately list our point-by-point responses. In the manuscript, all changes have been highlighted with “Track changes” option in Microsoft Word.

---

## [Decision Letter · Decision Letter 1]

7 May 2021

PONE-D-20-40689R1

Exploring the mechanism of hyperpermeability following glycocalyx degradation: Beyond the　glycocalyx as a structural barrier

PLOS ONE

Dear Dr. Iijima,

Thank you for submitting your manuscript to PLOS ONE. After careful consideration, we feel that it has merit but does not fully meet PLOS ONE’s publication criteria as it currently stands. Therefore, we invite you to submit a revised version of the manuscript that addresses the points raised during the review process.

Reviewer has specifically asked for the addition of a sentence explaining the limitations of the study in the Discussion. Otherwise the study has been amended and once this small modification is done, the paper is ready to be accepted.

We look forward to receiving your revised manuscript.

Kind regards,

Mária A. Deli, M.D., Ph.D.

Academic Editor

PLOS ONE

Journal Requirements:

Reviewers' comments:

Reviewer's Responses to Questions

**Comments to the Author**

1. If the authors have adequately addressed your comments raised in a previous round of review and you feel that this manuscript is now acceptable for publication, you may indicate that here to bypass the “Comments to the Author” section, enter your conflict of interest statement in the “Confidential to Editor” section, and submit your "Accept" recommendation.

Reviewer #1: All comments have been addressed

Reviewer #2: (No Response)

2. Is the manuscript technically sound, and do the data support the conclusions?

Reviewer #1: Yes

Reviewer #2: Partly

3. Has the statistical analysis been performed appropriately and rigorously? 

Reviewer #1: Yes

Reviewer #2: I Don't Know

4. Have the authors made all data underlying the findings in their manuscript fully available?

Reviewer #1: Yes

Reviewer #2: Yes

5. Is the manuscript presented in an intelligible fashion and written in standard English?

Reviewer #1: Yes

Reviewer #2: Yes

6. Review Comments to the Author

Reviewer #1: (No Response)

Reviewer #2: The main strength of the MS is the novel investigation of changes in endothelial cell expression of key regulators of the endothelial barrier after the glycocalyx is removed . However the methods to measure changes in barrier properties using very large dextran in mice and TEER in cultured endothelial monolayer limit the reliability of conclusion based on estimates of permeability. This weakness must be addressed in the Discussion. The following sentence must be included in the discussion to alert the reader to these limitations after line 361:

A particular issue is that dextrans as large as 2000K may not be useful probe molecules for more subtle, but still clinically important, changes in vascular permeability and measurements of TEER are insensitive to changes in glycocalyx permeability because the glycocalyx does not significantly restrict the small ion diffusion upon which this method is based.

7. PLOS authors have the option to publish the peer review history of their article (what does this mean?). If published, this will include your full peer review and any attached files.

Reviewer #1: No

Reviewer #2: No

---

## [Author Response · Author response to Decision Letter 1]

13 May 2021

Thank you for taking the time to review our work and provide constructive feedback.

We have added the limitations after line 361:

A particular issue is that dextrans as large as 2000K may not be useful probe molecules for more subtle, but still clinically important, changes in vascular permeability and measurements of TEER are insensitive to changes in glycocalyx permeability because the glycocalyx does not significantly restrict the small ion diffusion upon which this method is based.

---

## [Editor Report · Decision Letter 2]

17 May 2021

Exploring the mechanism of hyperpermeability following glycocalyx degradation: Beyond the　glycocalyx as a structural barrier

PONE-D-20-40689R2

Dear Dr. Iijima,

We’re pleased to inform you that your manuscript has been judged scientifically suitable for publication and will be formally accepted for publication once it meets all outstanding technical requirements.

Kind regards,

Mária A. Deli, M.D., Ph.D.

Academic Editor

PLOS ONE
---

## [Editor Report · Acceptance letter]

25 May 2021

PONE-D-20-40689R2 

Exploring the mechanism of hyperpermeability following glycocalyx degradation: Beyond the glycocalyx as a structural barrier 

Dear Dr. Iijima:

I'm pleased to inform you that your manuscript has been deemed suitable for publication in PLOS ONE. Congratulations! Your manuscript is now with our production department. 

Kind regards, 

on behalf of

Dr. Mária A. Deli 

Academic Editor

PLOS ONE